# Dynamics of Oxidative Stress in *Helicobacter pylori*-Positive Patients with Atrophic Body Gastritis and Various Stages of Gastric Cancer

**DOI:** 10.3390/diagnostics12051203

**Published:** 2022-05-11

**Authors:** Vladislav Vladimirovich Tsukanov, Olga Valentinovna Smirnova, Edward Vilyamovich Kasparov, Alexander Alexandrovich Sinyakov, Alexander Viktorovich Vasyutin, Julia Leongardovna Tonkikh, Mikhail Alexandrovich Cherepnin

**Affiliations:** Scientific Research Institute of Medical Problems of the North, Federal Research Centre “Krasnoyarsk Science Centre” of the Siberian Branch of Russian Academy of Science, 660022 Krasnoyarsk, Russia; ovsmirnova71@mail.ru (O.V.S.); clinic@impn.ru (E.V.K.); sinyakov.alekzandr@mail.ru (A.A.S.); alexander@kraslan.ru (A.V.V.); tjulia@bk.ru (J.L.T.); mikhail.cherepnin@yandex.ru (M.A.C.)

**Keywords:** atrophic gastritis, gastric cancer, oxidative stress, lipid peroxidation, antioxidant protection, *Helicobacter pylori*

## Abstract

Gastric cancer is a global health problem. The pathogenesis of this disease remains unclear. This study included 198 *H. pylori* (+) men aged 45 to 60 years old. Group A included 63 practically healthy men, group B included 45 men with severe atrophic body gastritis, group C included 37 men with epithelial gastric cancer stages I–II according to TNM, and group D included 54 men with epithelial gastric cancer stages III–IV according to the TNM scale. The content of malondialdehyde (MDA), diene conjugates (DCs), superoxide dismutase (SOD), catalase (CAT), glutathione S-transferase (GST), and glutathione peroxidase (GPO) was detected using an enzyme immunoassay (ELISA) or spectrophotometric methods in the blood plasma. The concentrations of MDA and DC were increased in the patients of group B compared with group A, and in patients of groups C and D compared with groups A and B. The ratio of MDA/SOD and MDA/CAT was decreased in the patients in group D compared with the patients in group C, and was significantly higher compared with group A. The ratios of MDA/GPO and MDA/GST increased linearly and were at a maximum in groups C and D. Our work determined that indicators of oxidative stress may be the biochemical substrate, which brings together the various stages of the Correa cascade, and may explain disease progression. The dynamics of changes in the content of SOD and CAT in the plasma in patients with gastric cancer may be a target of future investigations.

## 1. Introduction

In 2018, more than 1 million new cases of gastric cancer (GC) were detected worldwide and 782,683 deaths from this disease were registered [1]. The global age-standardized incidence of gastric cancer is 11.1 per 100,000 people, and the mortality rate is 8.2 per 100,000 people [2]. In Russia, the incidence of gastric cancer is high and varies in different regions, and is around 20–25 per 100,000 population [3]. The leading paradigm for the pathogenesis of gastric cancer is the Correa cascade, which postulates that *Helicobacter pylori* (*H. pylori*) infection causes the development of gastritis, which, under certain conditions, transforms into atrophy, metaplasia, dysplasia, and gastric adenocarcinoma [4,5]. Regarding this, in 2019–2021, European [6], British [7], and American [8,9] recommendations for the management of patients with precancerous changes in the stomach were published.

Currently, there are a number of works confirming the significance of oxidative stress regarding the development of gastritis and gastric cancer [10,11,12]. *H. pylori* disrupts intracellular processes in the epithelium, causing inflammation and epigenetic modifications [13,14]. Regarding inflammation, the response of the host organism is accompanied by the involvement of immune cells through their release of cytokines with the possible formation of reactive oxygen species (ROS), which leads to oxidative stress and contributes to deoxyribonucleic acid (DNA) damage and carcinogenesis [15]. In *H. pylori*-infected cells, processes that prevent the activation of the immune and antioxidant systems of the host organism occur [16]. Modern works have emphasized the prospects of studying oxidative stress indicators, the results of which may influence the development of new approaches for the prevention of gastric cancer [17,18,19].

### Aim of Research

The aim of this research was to study the indicators of lipid peroxidation and antioxidant enzymes in the blood plasma of *H. pylori*-positive men with atrophic gastritis and gastric cancer.

## 2. Materials and Methods

We examined 198 *H. pylori*-positive male patients aged 45 to 60 years old, among which 63 were practically healthy men (group A), 45 were patients with atrophic body gastritis (group B), 37 were patients with epithelial gastric cancer stages I–II according to the international TNM classification (group C), and 53 were patients with epithelial gastric cancer stages III–IV according to the international TNM classification (group D).

Group A included 63 men with a mean age of 48.7 ± 3.9 years old, without gastroenterological complaints and gastroenterological anamnesis. The recruitment of patients was carried out from persons who underwent a planned medical examination at the Research Institute of Medical Problems of the North, FRC KSC SB RAS. During medical monitoring, patients underwent clinical and biochemical blood tests; X-ray diagnostics; electrocardiography; and routine consultations with a cardiologist, and, if indicated, other specialists. Patients were included in group A only in the absence of severe chronic diseases of various organs and systems. To exclude atrophic body gastritis, the content of pepsinogen-1 and pepsinogen-2 was determined using an enzyme immunoassay using an ELISA analyzer “MULTISKAN FC” (Thermo Fisher Scientific, Waltham, MA, USA) using the test system “Gastropanel” (Biohit, Helsinki, Finland). The exclusion criteria for atrophic body gastritis were the content of pepsinogen-1 in the blood serum being more than 50 µg/L and the ratio of pepsinogen-1/pepsinogen-2 being more than 3.

Group B had 45 male patients (mean age 51.2 ± 4.9 years old) with chronic atrophic body gastritis (CAG). For the diagnosis of atrophic body gastritis, a morphological study of the biopsy specimens from three sections of the stomach obtained during fibroesophagogastroduodenoscopy (FEGDS) was used in accordance with the modified Sydney classification [20]. Group B included only patients with severe atrophic body gastritis.

Group C included 37 patients with stages I–II epithelial gastric cancer (mean age 51.2 ± 4.9 years) according to the international TNM classification. Group D included 53 patients with stages III–IV epithelial gastric cancer (mean age 53.2 ± 4.9 years old) according to the international TNM classification [21]. The diagnosis of gastric cancer was established by examining patients in the Krasnoyarsk Regional Oncological Dispensary on the basis of comprehensive clinical and instrumental examination, which included mandatory histological diagnostics.

In all of the examined groups, the determination of *H. pylori* was performed using a ^13^C-urease breath test using the Iris Doc apparatus. Only *H. pylori*-positive patients were included in the study.

The study material was venous blood, which was taken from patients in the morning, after a 12 h fast, from the cubital vein into Vacutainer tubes with sodium heparin solution (5 IU/mL). The blood plasma was isolated by centrifugation and was frozen at −80 °C for the biochemical analyses.

The concentrations of malonic dialdehyde (MDA), superoxide dismutase (SOD), catalase (CAT), glutathione S-transferase (GST), and glutathione peroxidase (GPO) were measured using the enzyme immunoassay method (ELISA) on Thermo Scientific Multiskan FC (Thermo Fisher Scientific, Waltham, MA, USA) apparatus using a human ELISA kit (SunRed Biotechnology Company, Shanghai, China) based on sandwich-linked ELISA using two specific and high affinity monoclonal antibodies. The concentration of studied substances was measured in ng/mL.

The ratios of MDA/SOD, MDA/CAT, MDA/GPO, and MDA/GST were calculated.

The content of conjugated diene structures (DCs) of lipid hydroperoxides was determined by spectrophotometry on a GENESYS 10S UV–VIS (Thermo Fisher Scientific, USA) apparatus, with measurements at a wavelength of 232 nm. The DC content was measured in µmol/L [22].

The study was conducted with permission from the Ethics Committee of the Scientific Research Institute of Medical Problems of the North, FRC KSC SB RAS (protocol no. 11 dated 11 November 2013). For working with the examined patients, the ethical principles required by the Russian Federation Constitution and the Declaration of Helsinki from the World Medical Association were met. All participants were informed of the purpose and procedure of the study and were asked to provide written informed consent to the study confirming their voluntary participation in the study.

### Statistical Analysis

Statistical data processing was carried out using the Statistica 7.0 software package (StatSoft, Tulsa, OK, USA). The analysis of the conformity of the type of distribution of the sign to the law of normal distribution was carried out using the Shapiro–Wilk test. The median (Me) and interquartile range of percentiles (C_25_–C_75_) were calculated when describing the sample. The significance of differences between the indicators of independent samples was assessed using the Mann–Whitney U test for pairwise comparison of groups and the Kruskal–Wallis H test to compare multiple groups (*p* < 0.05).

## 3. Results

We analyzed the gender, age, anthropometric, anamnestic characteristics, and laboratory parameters of the examined patients. Patients were gender identical (only males were examined) and practically did not differ in age. A decrease in body weight and an increase in the content of white blood cells in the blood were found in both groups of patients with gastric cancer in comparison with healthy individuals and persons with atrophic gastritis. Hemoglobin and erythrocytes decreased, and the neutrophil-to-lymphocyte ratio increased in patients with stages III–IV gastric cancer compared with healthy individuals and persons with atrophic gastritis. With the increase in stage of gastric cancer, the frequency of tobacco smoking increased and the frequency of alcohol consumption decreased (Table 1).

The content of lipid peroxidation products in the blood plasma was significantly increased in patients with gastric cancer in comparison with the control group, which consisted of practically healthy individuals, and with the atrophic gastritis group. For example, the level of MDA in the plasma of patients with stages III–IV gastric cancer was 72.7 times higher, and DC was 7.7 times higher in comparison with group A. It is noteworthy that in patients with severe atrophic body gastritis, a significant increase in both the primary and end products of blood lipid peroxidation was also determined in comparison with group A (Figure 1 and Figure 2).

The significance of differences was assessed by the Kruskal–Wallis H test and Mann–Whitney U test. Kruskal–Wallis H test: *p* < 0.001.

Mann–Whitney U test: p_1–2_ = 0.03; p_2–3_ = 0.03; p_3–4_ = 0.02.

The significance of differences was assessed using the Kruskal–Wallis H test and Mann–Whitney U test. Kruskal–Wallis H test: *p* < 0.001.

Mann–Whitney U test: p_1–2_ = 0.02; p_2–3_ = 0.001; p_3–4_ = 0.02.

All of the indicators of antioxidant protection we studied could be divided into the following two units: the main bifunctional system (SOD and CAT) and the glutathione link (GST and GPO). In patients with CAG and GC stages I–II, unidirectional increases in the concentration of superoxide dismutase and catalase enzymes relative to the control were detected. In patients with gastric cancer stages III–IV, differences were found in the form of a slight increase in the concentration of SOD and a severe increase in the content of CAT (Table 2). Enzymes of the glutathione link also showed their activity in different ways in the groups; there was an increase in the content of GST and GPO in patients with CAG and GC stages I–II, and an unbalanced activity in patients with GC stages III–IV in the form of a significant decrease in GST, and a slight increase in the concentration of GPO.

To assess the dynamics of the relationship between lipid peroxidation and antioxidant protection factors, we studied the relationship of MDA and the studied antioxidant enzymes. The patterns obtained were not unidirectional. The ratio of MDA/SOD and MDA/CAT clearly decreased in the patients in group D compared with the patients in group C, and were significantly higher in comparison with group A. This indicated that in patients with stages III–IV gastric cancer, the dynamics of an increase in SOD and CAT in the plasma was higher in comparison with the change in the concentration of MDA than in patients with stages I–II gastric cancer (Figure 3).

The ratios of MDA/GPO and MDA/GST had simpler and more obvious linear characteristics. The MDA/GST ratio was significantly increased in group D compared with groups A, B, and C. Significant differences in the MDA/GPO ratio were recorded when comparing group D with group A and group B (Figure 4).

## 4. Discussion

Oxidative stress is one of the significant aspects of the pathogenesis of diabetes, atherosclerosis, neurodegenerative diseases, and cancer of various localizations [23,24,25]. Reactive oxygen species (ROS), which include superoxide (•O_2_^−^), hydrogen peroxide (H_2_O_2_), and singlet oxygen and hydroxyl radical (•OH), are generally considered to be strong oxidizing agents and can accumulate in the organism when exposed to ultraviolet radiation, radiation exposure, over nutrition, and tobacco smoking [26]. The accumulation of ROS leads to the oxidation of biological macromolecules, which include proteins, lipids, and nucleic acids, causing their structural and functional changes. The oxidation of fatty acids leads to the formation of aldehydes. Currently, the most studied fatty acid is MDA [27]. MDA plays an essential role in cellular processes and can promote DNA changes, which is one of the stages of carcinogenesis [28,29,30].

The organism has a mechanism to protect it from oxidative stress and lipid peroxidation products. The main enzymatic antioxidants are superoxide dismutase (SOD), catalase (CAT), glutathione peroxidase (GPO), and glutathione S-transferase (GST). SOD catalyzes the dismutation of superoxide into oxygen and hydrogen peroxide. There are works that show that a decrease in the activity of SOD in the intestine causes stomach ulcers, and an increase in the activity of SOD is associated with the healing of ulcerative defects [31].

It has been shown that in the tissues of adenocarcinoma of the stomach and squamous cell carcinomas of the esophagus, there is an increased expression of manganese SOD compared with the normal mucosa [32]. Regarding the issue of changes in SOD activity during carcinogenesis, there is no single point of view. Thus, a decrease in SOD activity in the tumor tissue compared with healthy tissue has been observed in non-small-cell lung cancer [33], bladder cancer [34], and ovarian cancer [35]. An increase in the activity of this enzyme has been reported in colorectal cancer in tumor tissue [36] and blood serum [37].

CAT is a cellular enzyme that dismutates H_2_O_2_ into H_2_O and O_2_ [38]. *H. pylori* is capable of synthesizing CAT for deactivate H_2_O_2_ and surviving in the host organism [39]. A decrease in the activity of CAT has been recorded in patients with gastric adenocarcinoma [40].

GPO converts glutathione (GSH) to its oxidized form (GS-SG), and during this process, reduces H_2_O_2_ to H_2_O. It has been argued that GPO provides the first line of defense against reactive oxygen species in the gut [41]. GST belongs to the family of metabolic isoenzymes and has the ability to catalyze the conjugation of GSH with xenobiotic substrates for the purpose of detoxification [42,43].

A number of studies have been devoted to the role of oxidative stress and *H. pylori* in the pathogenesis of gastric cancer [17,29,44,45]. *H. pylori* virulence factors activate the signaling pathways of oxidative stress and are mediators of chronic inflammation in host cells. *H. pylori* is able to block the antibacterial effect of neutrophils and induce DNA methylation, which stimulates cell proliferation towards poorly differentiated gastric cells [12]. *H. pylori* strains containing CagA and VacA are characterized by an increased survival and adhesion ability, which increases damage to gastric epithelial cells and makes it possible to evade the immune response of the macro-organism [46]. It is known that CagA strains, as a result of the expression of spermine oxidase (SMO), induce higher levels of hydrogen peroxide (H_2_O_2_) production, which activate caspase-mediated epitheliocyte apoptosis and oxidative DNA damage [47,48]. CagA positivity is largely associated with an increased expression of tumor necrosis factor α (TNFα) and interleukin (IL)-8, both of which are markers of inflammation and oxidative stress [49]. Vacuolating cytotoxin A (VacA) induces Ca^2+^ influx, production of reactive oxygen species (ROS), activation of the nuclear factor kappa light chain enhancer of B cell activation (NFκB), and increased expression of chemokines, presumably increasing the influx of pro-inflammatory immune system cells to the site of bacterial infection of *H. pylori* [50].

Vacuolating cytotoxin A (VacA) induces Ca^2+^ influx, the production of reactive oxygen species (ROS), activation of the nuclear factor of kappa light chain enhancer of B cell activation (NFκB), and increased expression of chemokines, presumably increasing the influx of pro-inflammatory cells of the immune system to the site of *H. pylori* infection [50]. Intracellular Ca^2+^ and ROS of granulocytes further lead to oxidative stress [51,52]. VacA disrupts autophagy in human gastric epithelial cells and increases cellular ROS, which initiate the creation of an environment conducive to oxidative DNA damage with subsequent neoplastic transformations and cancer development [53,54].

An additional virulence factor of *H. pylori*, γ-glutamyltransferase (GGT), induces the production of H_2_O_2_ in gastric epithelial cells, and increases NFκB activation and the production of IL-8, a cytokine with pro-oncogenic properties [55,56,57]. Urease and neutrophil-activating protein A (NapA) are two virulence factors of *H. pylori* that promote neutrophil infiltration and ROS production [58]. Infiltrating neutrophils are characterized by increased resistance to apoptosis, have an extended lifespan, and are potential factors contributing to chronic inflammation caused by *H. pylori* infection [58,59,60]. NapA also has the function of attracting neutrophils to the site of infection and induces a respiratory burst of infiltrating neutrophils, which contributes to cellular oxidative stress and DNA damage. Hypothetically, neutrophil ROS should nonspecifically kill both infected and uninfected cells, as well as the pathogen. In addition, NapA protects *H. pylori* from the release of oxyradicals, maintaining the cycle of neutrophil infiltration and oxidative burst [61,62]. In macrophages, *H. pylori* also increases SMO expression and H_2_O_2_ release, which induces apoptosis in these cells [48].

Our results indicating a significant increase in lipid peroxidation in patients with stomach cancer are consistent with previously expressed views. We did not find information in the available literature that would allow us to clearly interpret the heterogeneous changes in antioxidant enzymes in patients with stages III–IV stomach cancer. One of the possible explanations for the severe increase in CAT in the blood of patients with stages III–IV stomach cancer may be that CAT belongs to intracellular enzymes, and metastatic stages of cancer are accompanied by massive cell cytolysis and the release of enzymes into the extracellular space.

## 5. Conclusions

This study used four groups of patients—practically healthy individuals, patients with severe atrophic body gastritis, and patients with gastric cancer of various stages according to the TNM classification—and led to several important results. Firstly, oxidative stress plays an important role in the pathogenesis of gastric cancer. Secondly, we demonstrated the presence of oxidative stress in the blood plasma of patients in group B in comparison with the patients in group A. From our point of view, these data confirm the reality of the Correa cascade. At the same time, indicators of oxidative stress can be a biochemical substrate that brings together different stages of the cascade and determines the progress of the disease. Third, another interesting pattern of our work was that we determined a severe increase in the concentrations of SOD and CAT in the blood plasma of patients with stages III–IV gastric cancer. In group D, the intensity of the increase in the content of SOD and CAT in the blood plasma was higher than the increase in the content of MDA. A similar pattern was not observed for glutathione enzymes. We hope that further research will be useful in explaining this phenomenon.

## Figures and Tables

**Figure 1 diagnostics-12-01203-f001:**
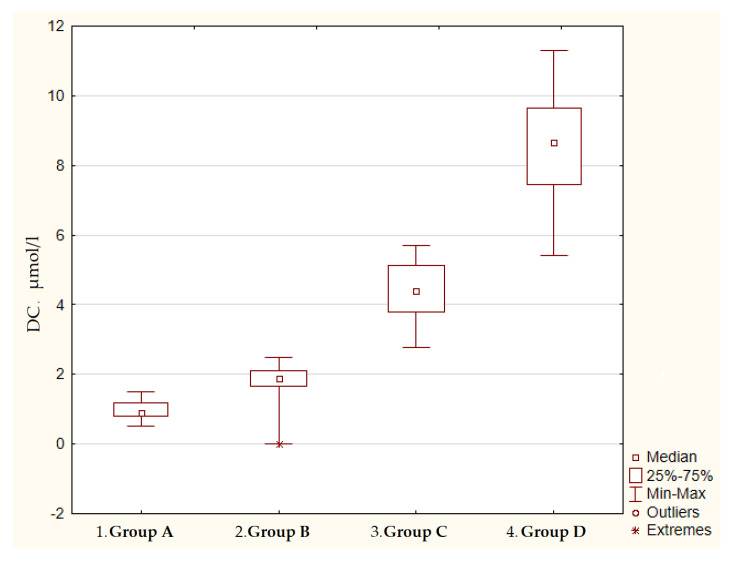
The content of diene conjugates in the blood plasma of patients with atrophic gastritis and gastric cancer.

**Figure 2 diagnostics-12-01203-f002:**
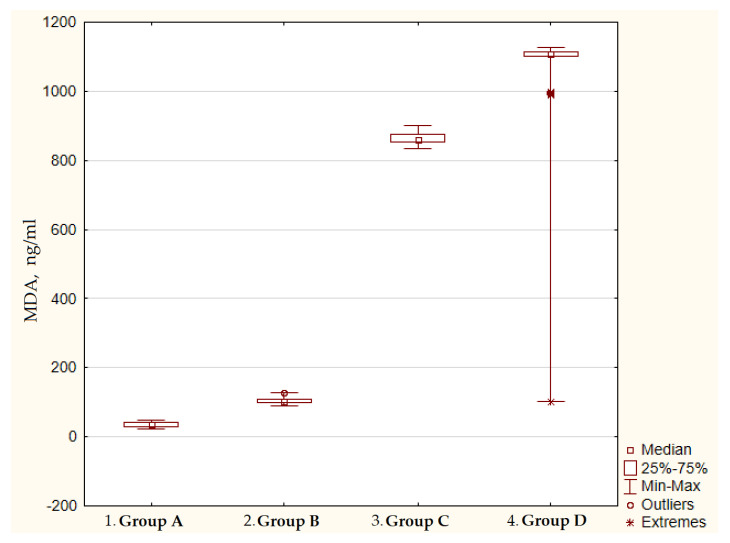
The content of malonic dialdehyde in the blood plasma of patients with atrophic gastritis and gastric cancer.

**Figure 3 diagnostics-12-01203-f003:**
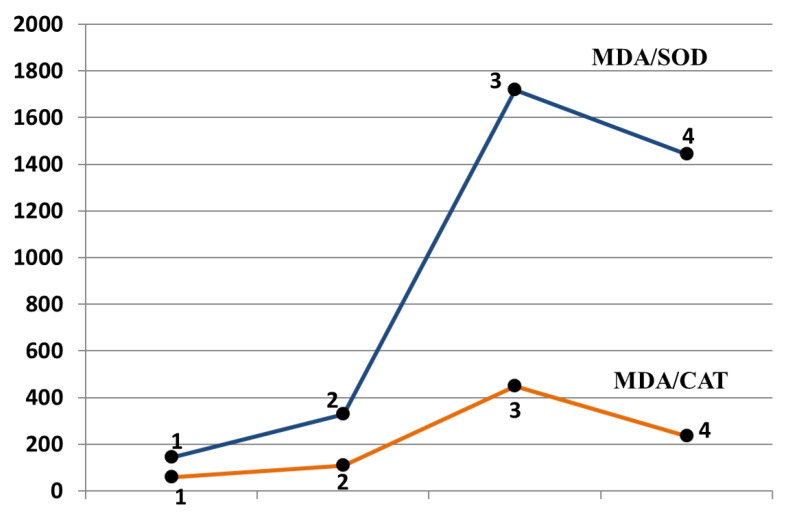
Concentration ratios of malonic dialdehyde to superoxide dismutase and malonic dialdehyde to catalase in the blood plasma of patients with atrophic gastritis and gastric cancer. 1—Group A; 2—Group B; 3—Group C; 4—Group D. The significance of differences was assessed by the Kruskal–Wallis H test and Mann–Whitney U test. Kruskal–Wallis H test: MDA/SOD *p* < 0.001; MDA/CAT *p* < 0.001. Mann–Whitney U test MDA/SOD: p_1–2_ = 0.03, p_2–3_ < 0.001, p_3–4_ = 0.02. Mann–Whitney U test MDA/CAT: p_1–2_ = 0.02, p_2–3_ < 0.001, p_3–4_ = 0.02.

**Figure 4 diagnostics-12-01203-f004:**
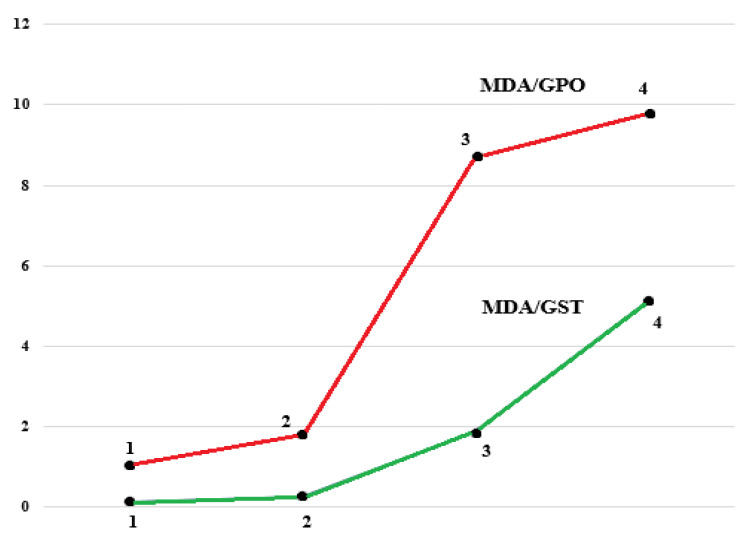
Concentration ratios of malonic dialdehyde to glutathione peroxidase and malonic dialdehyde to glutathione S-transferase in the blood plasma of patients with atrophic gastritis and gastric cancer. 1—Group A; 2—Group B; 3—Group C; 4—Group D. The significance of differences was assessed using the Mann–Whitney U test. Kruskal–Wallis H test: MDA/GST *p* < 0.001; MDA/GPO *p* < 0.001. Mann–Whitney U test MDA/GST: p_1–2_ = 0.03, p_2–3_ < 0.001, p_3–4_ < 0.02; Mann–Whitney U test MDA/GPO: p_1–2_ = 0.06, p_2–3_ < 0.001, p_3–4_ < 0.001.

**Table 1 diagnostics-12-01203-t001:** Anthropometric and anamnestic characteristics and laboratory parameters of patients with atrophic gastritis and gastric cancer.

Parameter	1. Control Group (n = 63) p	2. CAG (n = 45) p	GC Stages I–II (n = 37) p	GC Stages III–IV (n = 53) p
Men/Women n/n	63/0	45/0	37/0	53/0

Age. years Me (C_25_–C_75_)	48.9 (45.6–55.4)	51.1 (46.1–56.0)	51.4 (46.3–56.4)	53.0 (46.9–57.8)
		p_1–2_ = 0.8	p_1–3_ = 0.8	p_1–4_ = 0.6
			p_2–3_ > 0.9	p_2–4_ = 0.8
				p_3–4_ = 0.8
Height, cm Me (C_25_–C_75_)	179 (173–185)	178 (172–184)	176 (171–182)	177 (172–183)
		p_1–2_ = 0.9	p_1–3_ = 0.7	p_1–4_ = 0.8
			p_2–3_ = 0.8	p_2–4_ > 0.9
				p_3–4_ = 0.9
Weight, kg Me (C_25_–C_75_)	83.0 (74.5–91.0)	81.0 (72.0–90.5)	67.0 (59.5–75.0)	61.0 (54.5–68.0)
		p_1–2_ = 0.8	p_1–3_ = 0.003	p_1–4_ < 0.001
			p_2–3_ = 0.008	p_2–4_ < 0.001
				p_3–4_ = 0.1
Tobacco smoking Abs. (%)	12 (19.0%)	13 (28.9%)	13 (35.1%)	20 (37.7%)
		p_1–2_ = 0.3	p_1–3_ = 0.1	p_1–4_ = 0.04
			p_2–3_ = 0.7	p_2–4_ = 0.5
				p_3–4_ > 0.9
Alcohol consumption Abs. (%)	23 (36.5%)	18 (40.0%)	10 (27.0%)	10 (18.9%)
		p_1–2_ = 0.8	p_1–3_ = 0.4	p_1–4_ = 0.06
			p_2–3_ = 0.3	p_2–4_ = 0.04
				p_3–4_ = 0.5
Hemoglobin, g/dL Me (C_25_–C_75_)	14.1 (12.4–15.7)	13.1 (11.2–14.9)	11.7 (9.9–13.6)	10.4 (8.7–12.3)
		p_1–2_ = 0.4	p_1–3_ = 0.02	p_1–4_ = 0.005
			p_2–3_ = 0.2	p_2–4_ = 0.03
				p_3–4_ = 0.3
Red blood cells, ×10^12^/L Me (C_25_–C_75_)	4.7 (4.1–5.2)	4.5 (3.8–4.9)	4.1 (3.5–4.7)	3.7 (3.3–4.2)
		p_1–2_ = 0.6	p_1–3_ = 0.06	p_1–4_ = 0.01
			p_2–3_ = 0.3	p_2–4_ = 0.04
				p_3–4_ = 0.5
White blood cells, ×10^9^/L Me (C_25_–C_75_)	5.6 (4.4–6.9)	5.3 (4.1–6.7)	9.6 (7.1–11.7)	9.9 (7.3–12.1)
		p_1–2_ = 0.8	p_1–3_ = 0.005	p_1–4_ = 0.003
			p_2–3_ = 0.003	p_2–4_ < 0.001
				p_3–4_ = 0.7
Platelet, ×10^9^/L Me (C_25_–C_75_)	271 (216–338)	254 (194–319)	308 (176–444)	298 (165–413)
		p_1–2_ = 0.7	p_1–3_ = 0.2	p_1–4_ = 0.6
			p_2–3_ = 0.1	p_2–4_ = 0.4
				p_3–4_ = 0.8
Neutrophil-to-lymphocyte ratio Me (C_25_–C_75_)	2.21 (1.46–3.02)	1.98 (1.24–2.73)	2.86 (1.54–4.24)	3.15 (1.72–4.58)
		p_1–2_ = 0.8	p_1–3_ = 0.5	p_1–4_ = 0.04
			p_2–3_ = 0.1	p_2–4_ = 0.03
				p_3–4_ = 0.7

Note: the significance of differences in indicators was calculated using the Mann–Whitney U test and χ^2^.

**Table 2 diagnostics-12-01203-t002:** Indicators of antioxidant enzymes in the blood plasma in patients with atrophic gastritis and gastric cancer.

Groups Indicators	1. Control Group (n= 63) Me (C_25_–C_75_) p	2. CAG (n = 45) Me (C_25_–C_75_) p	GC Stages I–II (n = 37) Me (C_25_–C_75_) p	GC Stages III–IV (n = 53) Me (C_25_–C_75_) p
SOD, ng/mL	0.234 (0.225–0.252)	0.347 (0.295–0.386)	0.489 (0.512–0.605)	0.765 (0.754–0.805)
		p_1–2_ = 0.001	p_1–3_ = 0.03	p_1–4_ < 0.001
			p_2–3_ = 0.02	p_2–4_ < 0.001
				p_3–4_ = 0.02
CAT, ng/mL	0.472 (0.456–0.489)	0.987 (0.972–1.12)	1.86 (1.37–2.4)	4.52 (3.16–5.1)
		p_1–2_ = 0.03	p_1–3_ = 0.003	p_1–4_ < 0.001
			p_2–3_ = 0.01	p_2–4_ < 0.001
				p+ = 0.02
GST, ng/mL	241.3 (237.7–262.4)	384.2 (375.1–407.3)	448.6 (439.3–484.6)	217.6 (203.4–229.3)
		p_1–2_ = 0.01	p_1–3_ = 0.01	p_1–4_ = 0.01
			p_2–3_ = 0.002	p_2–4_ = 0.01
				p_3–4_ = 0.02
GPO, ng/mL	30.9 (21.4–62.9)	58.6 (42.1–76.5)	97.4 (78.1–104.4)	113.7 (105.2–131.7)
		p_1–2_ = 0.03	p_1–3_ = 0.001	p_1–4_ < 0.001
			p_2–3_ = 0.003	p_2–4_ < 0.001
				p_3–4_ = 0.02

Note: the significance of differences in indicators was calculated using the Mann–Whitney U test.

## Data Availability

Not applicable.

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
