# Peer review of "Dynamics of Oxidative Stress in Helicobacter pylori-Positive Patients with Atrophic Body Gastritis and Various Stages of Gastric Cancer"

_diagnostics, 2022, doi:10.3390/diagnostics12051203_

Round 1
Reviewer 1 Report
The article by Vladislav Vladimirovich Tsukanov and colleagues "Dynamics of oxidative stress in Helicobacter pylori-positive 2 patients with atrophic body gastritis and various stages of gastric cancer” it is well structured and thus the content of the research carried out is understood. The figures and tables are well described. In its entirety it is an article that can be published for such a crucial topic as oncogenesis in gastric pathology through H. pylori infection.
Author Response
Dear reviewer!
Thanks for your positive review.
Reviewer 2 Report
-the material and methods do not state whether they are in the group by chance
And they did EGDS so that it would be another confirmation that the respondents do not have it atrophic gastritis (only analysis of pepsinogen 1 and 2 was done)
-the results are too well written-I would keep only the numerical ones
facts without conclusions (last paragraph — perhaps included somewhere in
discussion)
-in the results I propose anthropometric and lab parameters of groups A and B.
who should be exposed to see how healthy they were respondents, groups C and D also
-discussion: 2 paragraph - whether SOD has been investigated in other tumors (no stomach only) - ref 31
-discussion whether there is a coincidence that a newer reference from ref 56 to confirm hypothesis of neutrophilic ROS
Author Response
- The material and methods do not state whether they are in the group by chance
And they did EGDS so that it would be another confirmation that the respondents do not have it atrophic gastritis (only analysis of pepsinogen 1 and 2 was done)
Dear reviewer, Institutional Ethics Committee of the Scientific Research Institute of Medical Problems of the North, FRC KSC SB RAS forbade us to perform EGDS with gastric biopsy in patients without gastroenterological complaints, and allowed us to determine atrophic gastritis by non-invasive methods. Screening for atrophic gastritis using pepsinogen-1, pepsinogen-2 and their ratio is recommended by international consensus such as MAPS II (Pimentel-Nunes P. et al., 2019; DOI: 10.1055/a-0859-1883) and Maastricht V ( Malfertheiner P. et al., 2017; DOI: 10.1136/gutjnl-2016-312288). We purposefully recruited a group of healthy patients during preventive planned medical examinations, which allowed us to conduct laboratory examinations.
- The results are too well written-I would keep only the numerical ones facts without conclusions (last paragraph — perhaps included somewhere indiscussion).
We agree with the remark, and in this regard, we remove conclusions from the results section. The second and last paragraphs in the “Results” have been removed.
- In the results I propose anthropometric and lab parameters of groups A and B,who should be exposed to see how healthy they were respondents, groups C and D also.
We have added a table containing anthropometric and lab parameters of groups A, B, C and D. Added the first paragraph in the "Results" section, added table 1. The table that was originally 1 was renamed to table 2.
- Discussion: 2 paragraph - whether SOD has been investigated in other tumors (no stomach only) - ref 31.
We agree with the remark, and in this regard, we have expanded the discussion on SOD in malignant tumors. An additional 5 literature references have been added (33-37) and revised bibliography.
- Discussion whether there is a coincidence that a newer reference from ref 56 to confirm hypothesis of neutrophilic ROS.
We agree with the remark and found a more recent article confirming the hypothesis from reference 56 (one more reference added to the bibliography).